# The Utilization of the WMO-1234 Guidance to Improve Citizen’s Wellness and Health: An Italian Perspective

**DOI:** 10.3390/ijerph192215056

**Published:** 2022-11-16

**Authors:** Letizia Cremonini, Marianna Nardino, Teodoro Georgiadis

**Affiliations:** Institute for the BioEconomy CNR, Via Gobetti 101, 40129 Bologna, Italy

**Keywords:** urban environment, integrated approach, health policies, ecosystem services, citizen’s health, regeneration policies

## Abstract

In 2019, the World Meteorological Organization published its “Guidance on Integrated Urban Hydrometeorological, Climate and Environment Services (Volume I: Concept and Methodology)” to assist WMO Members in developing and implementing the urban services that address the needs of city stakeholders in their countries. The guidance has relevant implications for not only protecting infrastructures from the impacts of climate change in the urban environment, but its proper declination strongly supports health-related policies to protect the population from direct and indirect impacts. Utilizing some principles of the guidance, the urbanized area of Bologna (Italy) was analyzed in order to furnish the municipality with tools coherent with the best practices actually emerging from the international bibliography to protect the citizens’ health of this city. Specifically, the analysis concentrated on the public spaces and the potential vulnerabilities of the fragile population to high-temperature regimes in the city. Utilizing the guidance as a methodological framework, the authors developed a methodology to define the microclimate vulnerabilities of the city and specific cards to assist the policymakers in city regeneration. Because the medieval structure of the city does not allow the application of a wide set of nature-based solutions, our main attention was placed on the possibility of furnishing the city with a great number of pocket parks obtainable from spaces actually dedicated to parking lots, thus introducing new green infrastructures in a highly deprived area in order to assure safety spaces for the fragile population.

## 1. Introduction

The urgency of actions across society, specifically in the urban system, is clearly outlined by Sustainable Development Goal 11 of the United Nations [1]. This specific goal, number 11, refers to “make cities and human settlements inclusive, safe, resilient and sustainable”. The word “safe” in the previous statement refers to the endangerment of life caused by the increase in extreme events and the strength of the concept of health expressed by WHO (World Health Organization). This concept refers to health as “a state of complete physical, mental and social well-being and not merely the absence of disease or infirmity”.

Nowadays, there is a massive body of evidence on how heat waves affect the health conditions of the general and fragile populations [2,3,4,5,6], specifically the elderly [7] (Journal of Geriatric Medicine and Gerontology), children [8,9], diabetic patients [10,11], and people affected by mental disorders [12,13]. These phenomena affect the population, increasing direct health risks and social isolation. We also have to bear in mind that we are still experiencing the effects of COVID-19, which significantly increased social isolation [14,15]; thus, there is an urgent need to spread best practices, especially for the Mediterranean regions where the worst effects are expected in a very short time [16,17,18].

The WHO 2011 report [19] found that many forms of asthma and allergies, as well as heart disease and strokes related to increasingly intense heat waves and cold spells, could be addressed by climate-friendlier housing measures. The report also notes that more attention should be paid to the housing risks of rapidly growing developing cities. Furthermore, for protecting populations, particularly vulnerable groups, the need to link knowledge relating to climate change and the response capacity of urban structures to this is becoming ever more immediate. For these reasons, close contact between different disciplines has been increasingly affirmed, which allows the development of integrated technologies and the practices of urban services [20,21].

The WMO guidance on Integrated Urban Hydrometeorological, Climate and Environmental Services [21] report the concepts and methodologies used by the public administration and stakeholders to closely link knowledge and actions. Only the understanding of the territorial reality which determines the urban and social structural fragility can allow the development of general strategies, which then identify the actions to be developed in the field. The document’s introduction explicitly references the SDG relating to sustainable cities and communities and the need to meet their needs. Specifically, to make the knowledge available for the development of services for the health-vulnerable population based on microclimate information at the city block scale, to advice on urban design, planning, and zoning. To further assist public administrations, the WMO has not limited itself to providing the theoretical and methodological basis to effect the change but has further strengthened the knowledge base by giving practical examples of cities that have already implemented these services as good practices [22]. Moreover, other authors from the WMO working group have produced an in-depth analysis of four case studies of cities located on different continents [23].

Once the methodology has been clearly traced, the public administrations are tasked with identifying fragility and vulnerabilities and implementing strategies and actions to protect the population. Therefore, it is necessary to take advantage of the potential that the various urban ecosystem services can express [24] and assess [25]. Specifically for Italy, an in-depth study on urban standards and ecosystem services was recently conducted and provided further indications of good practices in the Italian national territory [26]. This study is significant because it clarifies the evolution and innovations introduced in urban planning practices. Therefore, we have seen how it is possible to develop strategies and actions by combining them with the urban planning tools available from a methodological approach accompanied by exposure to good practices. This study aims to indicate how we can go to the level of detail of a single urban-architectural unit to implement strategies and actions at a micro-environmental level with the support of urban ecosystem services.

## 2. Methodology

The Municipality of Bologna, a city located in the Po Valley (Italy) and often subjected to the effects of heat waves, decided to implement its Adaptation Plan and, in particular, the General Urban Planning (PUG) tool with a study at the level of the entire city of climate vulnerabilities. The data collection and analysis methodologies are presented in Nardino et al. [27] and Nardino et al. [28] where a microclimate index and Urban Heatwave Thermal Index (UHTI), were developed. UHTI takes into account three principal urban remotely sensed elements: (a) surface materials represented by the LST (Land Surface Temperature), (b) vegetation represented by the Normalized Difference Vegetation Index (NDVI) and (c) urban morphology expressed in terms of SVF (Sky View Factor); their relationship with ground air temperature data is established through linear modeling. In particular, the relationship between air temperature and Vegetation Fraction Cover (VFC) was the following:T_VFC_ = −2.5 (VFC) + 25.79
indicating a decrease in air temperature when the vegetation is increasing.

This index, introduced in the General Urban Plan [29], now provides indications to construction companies and stakeholders on which values must be respected by operating in the different areas of the city.

Adopting that methodology already represents a basic level of protection for the population’s health. However, it remains of great importance to define how to implement specific protections according to the typology of the resident population, with the use of the places, and the specific characteristics of the micro architectural-urban structures. Therefore, the specific methodology suggested by the WMO guidance was applied, which, together with the physical analysis of the sites, accompanies a social evaluation of the use of the places to determine the best opportunities for choosing and applying ecosystem services.

The elements of the guidance considered in the study are reported in Figure 1. Even if the elements considered in the guidance are considerably higher than those used in the study, this limitation was made because the municipality mainly requests the urban response to heat waves.

The solutions identified by the Adaptation Plan are the actions and interventions based on a widespread application of ecosystem services represented by: urban parks, pocket parks, road trees, cool materials, and permeable floors. The decision to operate at this level was assumed considering that it is possible to intervene in the urban components (the streets, the squares, the green and water system, the buildings and materials present) that make up these paths as refreshment places where the fragile population can find relief from the thermal regime of the city. Thus, following the general frame indicated by the WMO guidance, the authors identified four pillars to assist policymakers in urban regeneration:The identification of the vulnerabilities of places and areas of aggregation for weaker groups of municipal property and properties of public interest.The detection of natural/environmental elements or particular conditions (including risk) in the weaker groups’ aggregation areas and, more generally, in the area.The selection of actions that mitigate the physiological climatic problem in the identified vulnerable points.Verification of the microclimate modeling of the selected project scenarios using microclimate modeling tools (ex-post simulations with ENVI-met [30]) regarding the specific objectives of each strategy.

In the study, five sensitive districts of Bologna were considered because the municipality had some concerns regarding the level of well-being of the resident population. However, applying the methodology to the entire city is very time-consuming. Therefore, the current main limitation to a punctual application of this method is the extensive calculation time necessary for the characterization of the different urban elements and the calculation of the well-being indices. However, this current limitation can be overcome by outsourcing the analysis to private companies or using cloud-computing techniques. Otherwise, it is possible to conduct only one analysis on specific frailties.

## 3. Results and Discussion

For the five sensitive city districts (Figure 2) specific cards were developed for this study to assist in the regeneration by the urban planners, the architects, the urban agronomists, and the social scientists. A category of cards was based on the neighborhood layout as morphology, the green and tree areas, and the ENVI-met-run simulations during a heat wave.

The second category of card details, on a smaller scale, was a single urban feature, such as a square or a street, where the fragilities that emerged in terms of well-being and the possible regeneration actions, served to increase resilience, in particular, that of health. Finally, the third and last category of cards defined the criticalities of individual urban elements and directly suggested the mitigation techniques applicable in that specific context, being also cross-checked with the public administration for both the feasibility of the proposed solution and the lack of collision with any other choices made during the planning phase.

Specifically, the results achieved for the Corticella district of Bologna are reported. This district is densely populated with a high resident population age, and some industrial infrastructures are in operation and disused. For these reasons, the district is given particular attention by the public administration.

Five vulnerabilities have been identified in the Corticella area to focus attention on (Figure 3), and the study has gone into a detailed scale, with related proposals for possible actions.

In Figure 3, vulnerabilities and the presence of places representing moments of social aggregation, which are of particular environmental sensitivity, are highlighted. For example, the presence of a school or a church immediately raises the problem of protecting vulnerable groups, represented in this case by children or the elderly. In these areas, particular attention must be paid to the issue of place regeneration. In fact, for zone 5, the school, a strong element of discomfort, generated presumably by local materials and the absence of vegetation, is evident from the microclimatic analysis.

In card 2 (Figure 4), there is an in-depth analysis of a vulnerability point that emerged for Corticella, already evidenced in card 1, to better understand the actions described and show what can be understood by safety paths. In this case, the card, as well as identifying vulnerability, indicates possible adaptation interventions for the problems highlighted. Therefore, the level two cards want to solve the issues related to the use of the city, such as the protection of the elderly in their compulsory travel to access services.

Once the microclimatic fragility of the specific environments has been identified, such as those reported in the cards proposed in the example, it is necessary to move on to the implementation phase of the regeneration through the application of NBSs. Unfortunately, historic cities, which in Europe represent a large part of the cultural and built heritage, are very rigid for a complete application of all available techniques. In particular, in structural terms, such as the blue and the gray, citizens often oppose those that involve a profound transformation because they require long construction periods with strong consequential impacts on urban mobility. Furthermore, the city regeneration with vast green infrastructures also encounters a certain level of resistance from the population due to the substantial reduction in parking spaces, the perceived safety, and the gentrification effects [31,32,33,34,35]. Although presenting these problems as a perception of one side of the citizenry, many techniques of active participation have been experimented with to ensure higher social acceptability of regeneration through vegetation, and also through the application of shared governance [36,37,38].

In the third card category, the analyses and proposals relating to the deeper level of urban regeneration are reported. The objective here is to solve specific exposures of the fragile population to a substantial risk due to the specificities of microscale urban planning. During structuring these actions, it is essential to take full advantage of all the urban features and components that structure the area in which the intervention is carried out. In the example (Figure 5), the goal is to create stopping and crossing points that guarantee a restorative “stop” or a “journey” for the weakest groups. Interventions near the pedestrian crossing can be assumed, generating shaded areas through the planting of trees and the generation of pocket parks, operating in some areas used for parking, and the desealing works. Some useful technical information about how to conduct a proper desealing is reported in the outcomes of the EU Life Project “SOS4Life” [39,40]. These small parks could also act as receptors for excess water in extreme rainfall. Suppose that there is a pre-existing garden in the proximity of the target site. In that case, its extension could be hypothesized, for example, by giving up portions of the sidewalk and immediately adjacent parking spaces to amplify the “urban forest” effect to the benefit of the immediately adjacent block.

The level of card 3, the deepest in the urban texture, allows for targeted substitutions in terms of urban decor and the development of spatial connections (routes for use by the elderly) protected in bioclimatic terms, and to correct errors of planning and site policies to better protect population health [41,42,43].

Even though the study was developed to be applied over the entire Bologna Municipality, the historical center of the city is characterized by a strong lack of vegetation coverage and a highly fragile population, mostly the elderly. Thus, specific attention was devoted to the application of NBSs in this part of the city.

In the broad overview of NBSs, which presents a high adaptive capacity in a rigid structure like the medieval city, and with a low divisive capacity on social issues, the pocket park is the most practical solution to respond to the theme of adaptation. Unlike large parks, where studies have allowed us to easily define the microclimatic effects of their presence [44,45,46,47], for pocket parks, there are no simple parameterizations capable of providing micrometeorological and bioclimatic effects. It is, therefore, necessary to proceed with the characterization of the ex-ante as per the proposed sheets for the evaluation of urban fragilities and then carry out, case by case, the ex-post runs of models such as ENVI-met to verify the project effects on the infrastructure. In general, it can be said that these effects are positive on the livability of the urban environment and allow the construction of bioclimatic safety paths for the fragile population [48,49,50].

The recent economic problems also increased the impact on the elderly population because the average low-income (the social pension is about 400–600 euros p.c.) forces people to shop at great distances to save money. Moreover, recent studies have demonstrated the inability of elderly and frail people to travel distances without appropriate recovery stops. In general, healthy subjects (i.e., those without functional alterations) can walk from 400 to 700 m in the time of the test (in which the proximity theories used for the realization of neighborhood services are found); a value below 400 m is already an indication of a poor functional capacity. For elderly and frail subjects, average values are considered to be those around 300–400 m in subjects with good functional capacity and less than 300 m in subjects with poor functional capacity. Furthermore, it was also assumed that these distances should be revised, as they are overestimated [51,52,53]. These distances are calculated without considering an unfavorable urban microclimate, such as during a heat wave, so a distance of fewer than 100 m acquires even more relevance.

In a city like Bologna, it could be important to estimate if the vulnerable population can reach several pocket parks to ensure bioclimatic protection. The pocket parks allow the creation of safety paths and facilitate access to essential services. For this reason, an analysis of the possible obtainable spaces for pocket parks from urban parking spaces was carried out.

The elderly population (+65) in Bologna is about 96,000 persons out of a total of 392,000 [54], and the number of young people (<14) is about 45,000. This means that the potential effects of a regeneration involve a large portion of the total resident population which is referred to as potentially fragile.

The way in which it is possible to obtain space to design pocket parks is just by suppressing several public parking spaces.

The number of parking spaces within the ancient medieval walls (Figure 6), corresponding to the city center, is around 10,000. A pocket park with sufficient space for rest and recreation for 3–4 persons is around 25-m squares, corresponding to the equivalent area covered by two parking spaces. Renouncing 10% of the spaces is reasonable without producing strong social opposition and makes available an area of 12,500 square meters for de-sealing, enough for about 500 pocket parks. Considering the total area of the center of Bologna, the pocket park density would be one every 8600 square meters. This surface area means that the response to the need for proximity is a pocket park planned for every 100–200 linear meters. Thus, the weaker sections of the population would be provided with a microclimatic security area to stop and regenerate. The existing public green infrastructures of the city center (in Figure 7, the green polygons that include grass and tree species) inside the walls today cover 381,000 square meters; assuming the addition of 12,500 square meters of pocket parks throughout the historic city, equally distributed, means a total of 393,500 sq.m. Pocket parks would therefore account for 3.3% of the total area of public parks, but their location would make the difference from a microclimatic point of view. Figure 7 represents the streets of the city within the walls in which the parking spaces have been registered, and using this as a first approximate representation, if each point is assumed as the creation of a pocket park, they represent the coverage that would be guaranteed from a point of view of the physiological well-being of the weaker groups.

With regards to the potential decrease in air temperature, applying the linear formulation proposed for the city of Bologna, with the vegetation introduced by the pocket parks, a value of less than 1% would be obtained. However, it is highly questionable to apply a parameterization obtained on the more massive presence of vegetation outside the historic center by directly adding the value of the scattered vegetation placeable inside the historic city walls, even if the parameterization is linear. However, the direct shading effects for the population that would benefit from the ecosystem service offered by this are certainly to be considered positive.

## 4. Conclusions

The application of the methodology, which has its roots in the WMO guidance, has shown that it can highlight urban criticalities with a potential impact on the physiological equilibrium of the population in general, but with particular regard for the vulnerable. Moreover, this methodology on the whole urban texture would allow the solving and reconciliation of spaces for the bioclimatically fragile and social use. In particular, the structure of the cards allows the public administration to ‘observe’ the city at different levels of complexity and to see if these levels harmonize with a typical design.

The different levels of description or in-depth analysis of the maps ensure a multi-role service to planners, allowing them first to grasp the vulnerabilities arranged over a large area or neighborhood and then deepen them at an urban matrix level, including factors relating to the accessibility of places in the protection of the physiological parameters of the weaker groups. Further still, in the deeper level of description, the ability presents itself to operate on the individual elements of the urban texture by proposing operational solutions.

Applying this methodology requires a good database of data and knowledge of the territory and its use by the population. It can therefore be foreseen that its possible application is preferably addressed to medium to large cities.

It was demonstrated that converting a limited number of parking plots (10%) into pocket parks can assure the fragile population with rest and safety spaces within the physiological indications of the international scientific bibliography.

The original outcome of this study indicates for European Medieval cities, a useful approach is to apply at least one adaptation tool, even if the specific “rigidity” of the architectural context does not allow deep infrastructural intervention.

As already highlighted, a problem for large cities relates to the calculation time to solve the individual urban characteristics that can hardly be solved directly by the public administration. However, the problem can be generally solved by externalizing the ENVI-met fluid dynamics model’s initialization and calculation.

## Figures and Tables

**Figure 1 ijerph-19-15056-f001:**
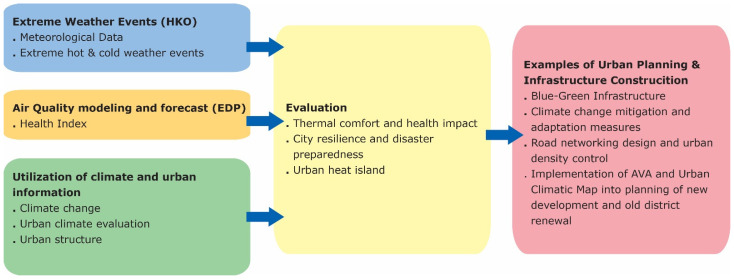
Elements of the WMO guidance [21] utilized for the present study.

**Figure 2 ijerph-19-15056-f002:**
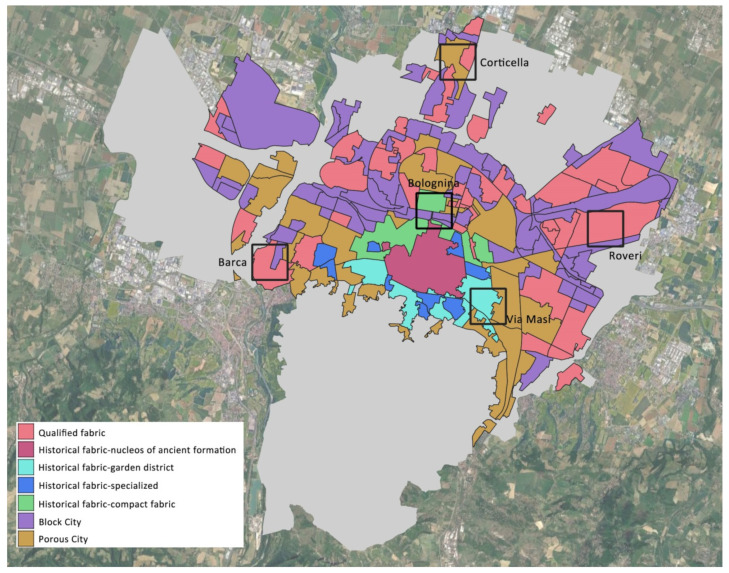
Map of the sensitive city districts of Bologna analyzed in the study.

**Figure 3 ijerph-19-15056-f003:**
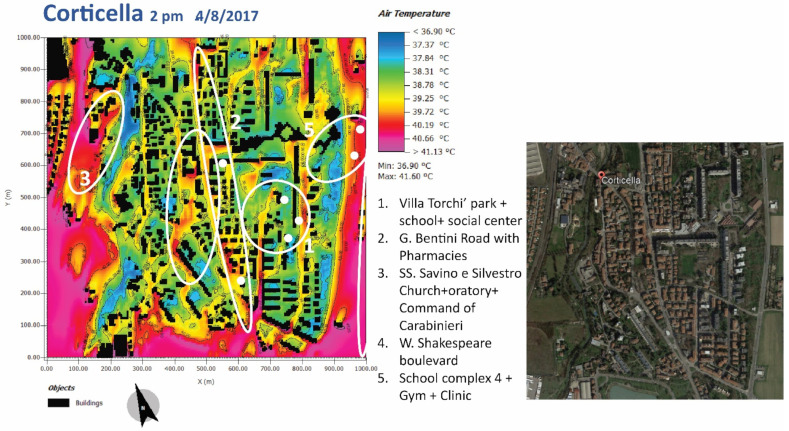
Card 1: Identification of vulnerabilities in the map of the air temperature at 1.8 m at 2 p.m. simulated in the Corticella area and the ortho-photo taken from Google Maps updated to 2019.

**Figure 4 ijerph-19-15056-f004:**
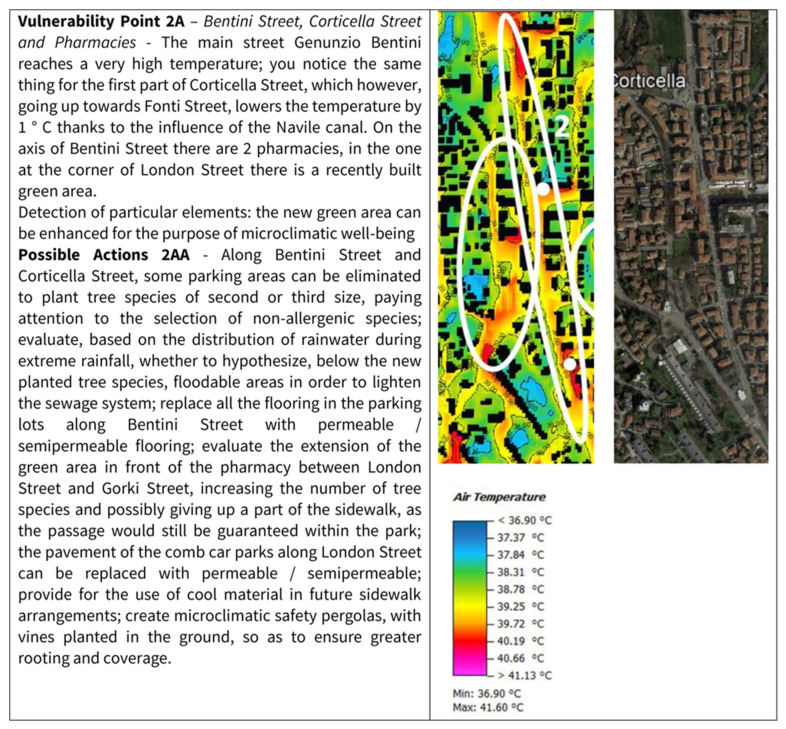
Card 2: Description of Corticella-specific vulnerability corresponding to Point 2 of the previous card, with possible related actions.

**Figure 5 ijerph-19-15056-f005:**
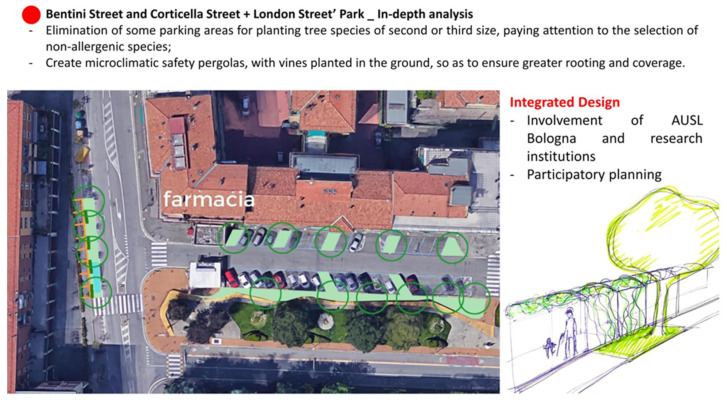
Card 3: More detailed level of intervention in the urban structure according to the use of space. The proposed crossroads were analyzed for the presence of a Pharmacy (Farmacia) representing a hot spot for the fragile population. On the right is a possible regeneration solution for the site. While on the left is an indication to the municipality for the reduction in parking lots to be substituted by small green areas.

**Figure 6 ijerph-19-15056-f006:**
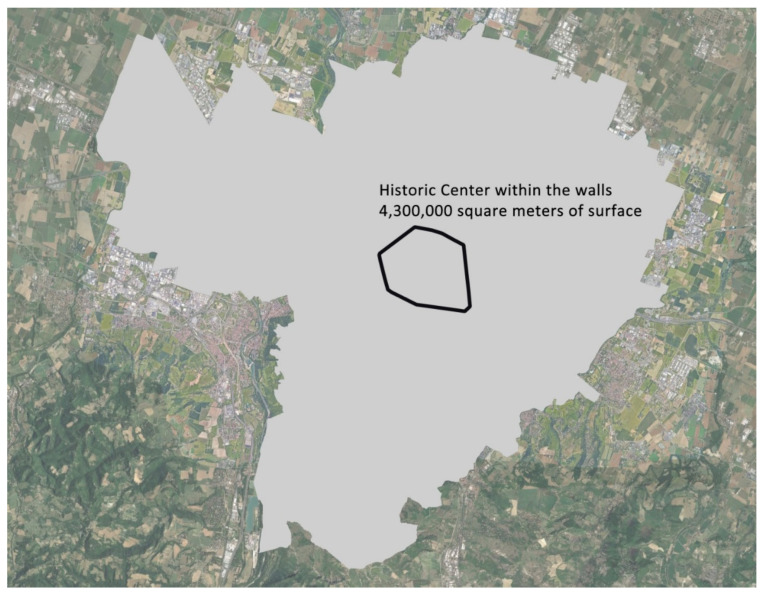
Location of the historical city walls with an included area of 4,300,000 square meters.

**Figure 7 ijerph-19-15056-f007:**
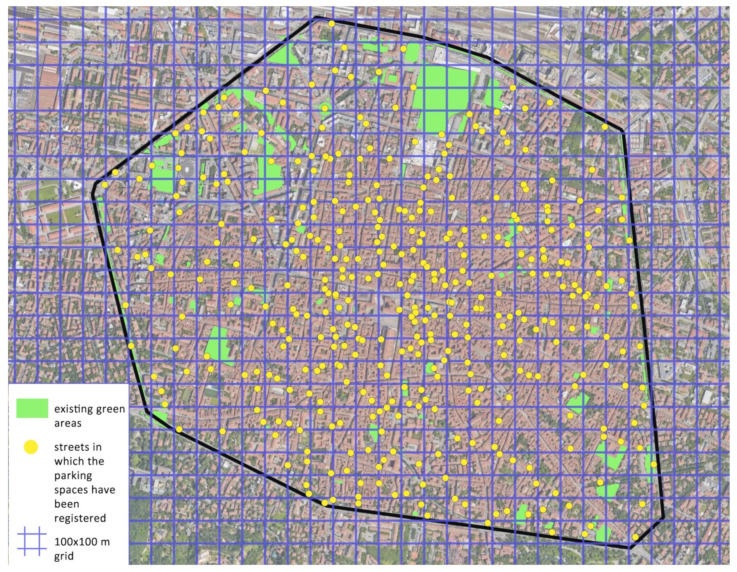
The map of the historical center inside the walls highlights the existing green areas in green and the streets for which parking spaces have been registered in yellow. The blue grid has a mesh of 100 × 100 m and aims to bring out the potential impact of pocket parks on the stiffer historical fabric of the municipal area.

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
