# Peer review of "The Utilization of the WMO-1234 Guidance to Improve Citizen’s Wellness and Health: An Italian Perspective"

_ijerph, 2022, doi:10.3390/ijerph192215056_

Round 1

Reviewer 1 Report

The article is interesting, combining theory and practice, and gives pointers to concrete actions for adapting cities to climate change and protecting vulnerable groups of residents to increasingly frequent and longer-lasting heat waves.

Unfortunately, after reading the article, one gets the impression that it is reproductive and describes actions taken by the Bologna authorities. It is not mentioned in any part of the paper what the original contribution of the author(s) is. There is a lack of information about data sources, and the methodology described does not indicate the stages of the research procedure carried out by the author(s). Perhaps this impression is misleading. If this is indeed the case, the text needs to be reworded so that it is obvious what the author/authors' contribution to the results of the presented research is.

In addition, it is worth clarifying:

- what "Covenant of Majors" is referred to in the abstract,

- what population groups, other than children and the elderly, are considered more vulnerable to heat waves,

- what other measures can be taken to protect urban residents from heat waves?

The study is based on a limited number of publications.

Reviewer 2 Report

The article submitted for review, entitled "The utilization of the WMO-1234 Guidance to improve citizen's wellness and health: an Italian outcome", deals with scientific and technical issues describing the impact of natural phenomena and anthropogenic factors on the quality of the environment and our lives. Social and economic development adversely affect the state of the environment. Pollutant emissions adversely affect the quality of life of society. Any work aimed at improving the quality of the environment is therefore very valuable.
Unfortunately, however, I believe that every part of the work needs significant revision. Please see: Instructions for Authors.
Abstract
The abstract should be an objective representation of the article. Unfortunately, it does not contain information that encourages the viewer to read the entire article. Please clearly indicate what is new about the paper, what methods were used and what are the most important results.

Introduction
The literature review needs to be expanded. It does not discuss studies by other authors in the field. Their drawbacks or limitations are not pointed out. The scientific objective of the work is also missing. However, the general idea of the research (line 65-70) is very good.

Materials and Methods
I believe that presenting the method of the research in the form of a reference to other publications is not sufficient. If others have already done such studies, what will be their novelty? These types of analyses are time-consuming for large areas so they are done by private companies. It is difficult to call a scientific article a succession of studies carried out for individual cities without introducing an element of novelty. Of course, the object of the study can be original and will yield valuable results. Please indicate what guided you in selecting the object of study. If your paper is original, please indicate this in its content.

Results
Has this hypothesis been clarified? "These small parks could also act as receptors for excess water in extreme rainfall. Suppose there is a pre-existing garden in the proximity of the target site. In that case, its 185 extension could be hypothesised, for example, by giving up portions of the sidewalk and immediately adjacent parking spaces to amplify the 'urban forest' effect to the benefit of the immediately adjacent block."
The research idea is a very good one. I suggest showing what results we will get by making the proposed changes. To what extent will we reduce the temperature by replacing the parking space with grass? What should be the intensity of such investments to reduce temperature and improve water retention? How will we affect traffic problems in the city? There are many questions and no answers.

Discussion:
The paper does not interpret the results of our own research with previous studies.  No reference was made to the working hypotheses. No reference has been made to future research directions in this area.
Unfortunately, but I believe that in its current form, the article should not be published. It lacks an original approach to the scientific problem. Please undertake a critical review of the literature in this area. Show that there is a scientific problem. Only after pointing it out, select a research methodology and present concrete results.
As it stands, it is a report and not a scientific article.

Round 2

Reviewer 1 Report

After the changes made, the article is definitely better embedded in the literature, thus increasing its substantive value.

However, there are still minor shortcomings in the text that require the authors' intervention. They are as follows:

The lack of a clearly formulated aim of the study. Is one of the objectives methodological in nature?

It is still not entirely clear what is the author's contribution to the methodology presented in the study. The Methodology section says "Following the methodology proposed ..." (line 117), but it is not clear whether this methodology was proposed in the Guidelines, the Adaptation Plan or by the authors of the study. Are Charters 1, 2 and 3 an original idea of the authors? It seems that they are, but this would require clear emphasis in the text.

The spatial scope of the study became unclear after the introduction of additional content related to the historic center of the city.

The wording needs improvement:

- "These effects affect this population, producing exposure ..."(line 40) - effects not "producing exposure",

- "effects of COVID, which .... also worsened the very effects of the heat waves" (line 42) - effects not "worsened the very effects of the heat waves". Rather, they exacerbated the isolation,

- abbreviations are used whose full names are not explained in the text (e.g., LST - line 89; SVF - (line 91).

Author Response

1.The lack of a clearly formulated aim of the study. Is one of the objectives methodological in nature?

R: The abstract was changed accordingly.

2.It is still not entirely clear what is the author's contribution to the methodology presented in the study.

R: We have clarified in the text the authors contribution.

3.The Methodology section says "Following the methodology proposed ..." (line 117), but it is not clear whether this methodology was proposed in the Guidelines, the Adaptation Plan or by the authors of the study.

R: We changed the text to clarify this point.

4.Are Charters 1, 2 and 3 an original idea of the authors? It seems that they are, but this would require clear emphasis in the text.

R: Yes, we clearly reported the authors contribution.

5.The spatial scope of the study became unclear after the introduction of additional content related to the historic center of the city.

R: We specified that the introduction of this additional content regarded specifically the city center and the reasons why we devoted a specific attention to this issue.

6.The wording needs improvement:

- "These effects affect this population, producing exposure ..."(line 40) - effects not "producing exposure",

- "effects of COVID, which .... also worsened the very effects of the heat waves" (line 42) - effects not "worsened the very effects of the heat waves". Rather, they exacerbated the isolation,

- abbreviations are used whose full names are not explained in the text (e.g., LST - line 89; SVF - (line 91).

R: We changed the text accordingly.

Reviewer 2 Report

I accept the authors' correction of the paper.

Author Response

We really thank the reviewer for the very useful comments.